# Short-Term Effects of Low-Level Ambient Air NO_2_ on the Risk of Incident Stroke in Enshi City, China

**DOI:** 10.3390/ijerph19116683

**Published:** 2022-05-30

**Authors:** Zesheng Chen, Bin Wang, Yanlin Hu, Lan Dai, Yangming Liu, Jing Wang, Xueqin Cao, Yiming Wu, Ting Zhou, Xiuqing Cui, Tingming Shi

**Affiliations:** 1School of Public Health, Wuhan University, 115 Donghu Road, Wuhan 430071, China; zeshengchen2022@163.com (Z.C.); c448517577@163.com (X.C.); 15271936238@163.com (Y.W.); 2Hubei Provincial Key Laboratory for Applied Toxicology, Hubei Provincial Center for Disease Control and Prevention, 35 Zhuodaoquan North Road, Wuhan 430079, China; peggy20110906@163.com (Y.L.); danni-joy@163.com (J.W.); 3Enshi Center for Disease Control and Prevention, 5 Hongqi Avenue, Daqiao Road, Enshi 445000, China; w18671809903@163.com (B.W.); 18972415932@163.com (Y.H.); dai580150@163.com (L.D.); 4Hubei Province Key Laboratory of Occupational Hazard Identification and Control, Medical College, Wuhan University of Science and Technology, Wuhan 430065, China; zhouting84@wust.edu.cn

**Keywords:** nitrogen dioxide, stroke, risk of incidence, generalised additive model, time-series analysis

## Abstract

Previous studies found that exposure to ambient nitrogen dioxide (NO_2_) was associated with an increased risk of incident stroke, but few studies have been conducted for relatively low NO_2_ pollution areas. In this study, the short-term effects of NO_2_ on the risk of incident stroke in a relatively low-pollution area, Enshi city of Hubei Province, China, were investigated through time-series analysis. Daily air-pollution data, meteorological data, and stroke incidence data of residents in Enshi city from 1 January 2015 to 31 December 2018 were collected. A time-series analysis using a generalised additive model (GAM) based on Poisson distribution was applied to explore the short-term effects of low-level NO_2_ exposure on the risk of incident stroke and stroke subtypes, as well as possible age, sex, and seasonal differences behind the effects. In the GAM model, potential confounding factors, such as public holidays, day of the week, long-term trends, and meteorological factors (temperature and relative humidity), were controlled. A total of 9122 stroke incident cases were included during the study period. We found that NO_2_ had statistically significant effects on the incidence of stroke and ischemic stroke, estimated by excess risk (ER) of 0.37% (95% CI: 0.04–0.70%) and 0.58% (95% CI: 0.18–0.98%), respectively. For the cumulative lag effects, the NO_2_ still had a statistically significant effect on incident ischemic stroke, estimated by ER of 0.61% (95% CI: 0.01–1.21%). The two-pollutant model showed that the effects of NO_2_ on incident total stroke were still statistically significant after adjusting for other air pollutants (PM_2.5_, PM_10_, SO_2_, CO, and O_3_). In addition, the effects of NO_2_ exposure on incident stroke were statistically significant in elderly (ER = 0.75%; 95% CI: 0.11–1.40%), males (ER = 0.47%; 95% CI: 0.05–0.89%) and cold season (ER = 0.83%; 95% CI: 0.15–1.51%) subgroups. Our study showed that, as commonly observed in high-pollution areas, short-term exposure to low-level NO_2_ was associated with an increased risk of incident stroke, including ischemic stroke. Males and elderly people were more vulnerable to the effects of NO_2_, and the adverse effects might be promoted in the cold season.

## 1. Introduction

Stroke, also known as a cerebrovascular accident, is a severe threat to human health. The latest study of the Global Burden of Cardiovascular Disease, published in 2020, showed that stroke was the second-leading cause of death, and the stroke burden in terms of DALYs keepped increasing [1]. During the COVID-19 pandemic, stroke has remained one of the important causes of poor health worldwide [2]. According to the Chinese disease burden study in 2013, the top three death causes were stroke, ischemic heart disease, and chronic obstructive pulmonary disease, and there were approximately 1.1 million deaths from stroke in China in 2013 [3]. Stroke can be categorised into two major clinical types, ischemic and haemorrhagic stroke, which account for 85% and 15% of the cases, respectively. Stroke is a complex multifactorial disease affected by multiple factors. According to previous studies, various factors including air pollution, transportation noise, smoking, substance abuse, and physical inactivity might be contributing factors to stroke [4,5]. In recent decades, emerging studies demonstrated the important role of air pollutants on stroke morbidity and mortality [6,7,8]. Shin et al. found the associations between air pollution, including PM_2.5_, NO_2_, O_3_, and incidence of stroke even at relatively low concentrations [8]; a multicity study about acute effects of air pollutants on stroke mortality showed that exposure to NO_2_ and SO_2_ were positively associated with daily stroke mortality in eight Chinese cities [9]. A cohort study in Japan found that long-term exposure to air pollutant NO_2_ significantly increased the risk of stroke mortality [10]. However, limited studies have been performed to investigate the effects of NO_2_ on the risk of incident stroke [11,12,13].

NO_2_ is a typical gaseous ambient air pollutant from motor vehicles, fossil fuel plants, indoor gas stoves, and tobacco smoking [14,15]. A meta-analysis combining different countries and including more than 23 million participants confirmed that NO_2_ exposure could increase both the incidence and mortality risk of stroke [7]. This meta-analysis provided comprehensive evidence for the relationship between NO_2_ exposure and stroke, but its focus was mainly on areas with serious atmospheric pollution. The effects of NO_2_ on stroke risk in low-pollution areas, especially on the incidence of stroke, have not been well-studied. Considering the non-negligible population exposed to relatively lower atmosphere pollution worldwide, studying the effects of low-level NO_2_ on the risk of incident stroke is necessary.

In this study, we chose Enshi city, a city located in western Hubei Province, China, as the research area to explore the short-time effects of low-level NO_2_ on the risk of incident stroke. Compared with other cities in China, the ambient air NO_2_ concentrations in Enshi city were at relatively lower levels [16]. Subgroup analysis was conducted according to sex, age, and season to demonstrate susceptibility in different subgroups and seasons.

## 2. Materials and Methods

### 2.1. Data Sources

All the incident stroke data were derived from the China Disease Prevention and Control Information System [17]. The stroke incidence cases were clinically diagnosed by the local sentinel hospitals and were simultaneously reported to the China Disease Prevention and Control Information System through case report cards. Through the system, we downloaded the data of all reported incident stroke cases, including the ID number, sex, age, date of birth, onset data of stroke, type of stroke diagnosis, ICD-10 encoding, and current address for each case. According to the International Classification of Diseases version 10 (ICD-10) (World Health Organization, Geneva, Switzerland), stroke was divided into different subtypes (haemorrhagic stroke code: I60–I62; ischemic stroke code: I63; other types of stroke codes: I64). Subgroup analysis was based on sex, age, and season. According to World Health Organisation (WHO) standards, the elderly group was defined as age ≥ 65 years, and the non-elderly group was defined as age < 65 years [18]. Seasons were divided into the cold season (October, November, January, December, January, February, and March) and the warm season (April, May, June, July, August, and September) according to local climatic characteristics [19].

The daily air pollutant data including PM_2.5_, PM_10_, SO_2_, NO_2_, CO, and O_3_ were collected from two national air quality control automatic monitoring stations (Figure 1), the Enshi Electric Power Corporation monitoring station and Hubei University for Nationalities monitoring station, from 1 January 2015 to 31 December 2018 [20]. According to the WHO’s Air Quality Guidelines, we used the daily maximum of 8 h means for ozone and daily means for the other pollutants. Missing values in air-pollution data were imputed using the multiple-interpolation method [21]. The meteorological data, including daily average temperature (temp) and relative humidity (rhum) of the same period, were collected from Enshi Meteorological Bureau.

### 2.2. Statistical Analysis

The descriptive statistics including mean (Mean ± SD), minimum (Min), maximum (Max), and quartile (P_25_, Median, P_75_) were used to describe the data on incident stroke cases, air pollutants, and meteorological factors. Spearman rank correlation analysis was used to analyse the correlation between air pollutants and meteorological factors. In a time-series study, the generalised additive model (GAM) analysis is usually used to evaluate the risk of short-term air pollutant exposure, including pollutants with relatively low concentrations [11,19,22]. A GAM is the classical addendum of general linear models, which provides a structure for generalising a general linear model by allowing additivity of non-linear functions of the variables [23]. In this study, the daily incidence of stroke in Enshi residents was a somewhat rare event, and the correlation between explanatory variables and the number of stroke incidences per day were mainly non-linear. Therefore, the GAM based on Poisson distribution was used to explore the effects of NO_2_ on the risk of incident stroke, in which potential confounding factors, including public holidays (Holiday), day of the week (DOW), long-term trends (Time) and meteorological factors (temp and rhum), were controlled [24,25,26]. In the GAM model, we adjusted for the effects of long-term trends and meteorological factors on stroke by natural cubic spline function (*ns*). DOW and Holiday were dummy variables that reflect week and holiday effects. To verify the independent effects of NO_2_ on stroke, we established two-pollutant models, in which the interference effects of PM_2.5_, PM_10_, SO_2_, CO, and O_3_ were adjusted, respectively.

The equation for single-pollutant model was as follows:Log [E (Yt)] = α + β Xt + *ns* (Time, *df* = 7/per year) + *ns* (temp, 𝑑𝑓 = 3) + *ns* (rhum, 𝑑𝑓 = 3) + DOW + Holiday(1)

The equation for two-pollutant model is as follows:Log [E (Yt)] = α + β Xt + β X_1_t + *ns* (Time, *df* = 7/per year) + *𝑛**s* (temp, 𝑑𝑓 = 3) + *ns* (rhum, 𝑑𝑓 = 3) + DOW + Holiday(2)

In the above equations, E (Yt) is the expected number of daily incident strokes on “t” day. Yt is the actual number of daily incident strokes on “t” day. The Greek letter α is the intercept of the equation. The Greek letter β stands for the regression coefficient that indicates the relative risk of stroke associated with one-unit increase in NO_2_ concentration. Xt represents NO_2_ concentration on “t” day. X_1_t represents other air pollutants (PM_2.5_, PM_10_, SO_2_, CO, or O_3_) concentration on “t” day; *df* is the degree of freedom of each parameter. Based on the experience of previous large-scale studies [24,25,26] and the minimum principle of the Akaike Information Criterion (AIC), the *df* of temp, rhum and time are 3, 3, and 7, respectively [27].

We also estimated the lag effects of low-level NO_2_ on the risk of incident stroke, including single-day lag effects (lag0, lag1, lag2, lag3, lag4, lag5, lag6, and lag7) and cumulative lag effects (lag01, lag02, lag03, lag04, lag05, lag06, and lag07). The lag effects referred to the effect of the air-pollution concentration on the previous day on a health outcome on the day. In single-day exposure models, the lag0 corresponded to the current-day air-pollutant concentration, and lag1 referred to the average concentration of air pollutants on the day before stroke incidence, etc. In cumulative exposure models, the lag01 corresponded to the 2-day moving average concentration of air pollutants on the present day and previous day, etc. We used excess risk (ER) to represent the percentage increase in the daily incidence of stroke associated with one unit (1 μg/m^3^) increase in NO_2_, and the formula was ER = [exp(β) − 1] × 100%.

Sensitivity analysis was conducted to check the stability of the results including (1) using different *df* values for long-term trends (7–9) and meteorological factors (3–6) in the GAM model and (2) adding other air pollutants (PM_2.5_, PM_10_, SO_2_, CO, and O_3_) to the single-pollutant model to create two-pollutant models. If similar results were observed in the sensitivity analyses, the results were stable.

The main analytical tool used in this study was the mgcv software package in R software (version 3.5.2) (R development Core Team, Vienna, Austria).

## 3. Results

### 3.1. Descriptive Analysis and Correlation Analysis

Table 1 presents the results of descriptive statistics for daily incident stroke cases, air-pollutant concentrations, and meteorological factors in Enshi city, 2015–2018. During 1 January 2015 and 31 December 2018, there were a total of 9122 incident stroke cases in Enshi city, and the annual size of the incident stroke cases ranged from 1448 to 2683 (Appendix A). Of the 9122 cases, the cases attributed to ischemic stroke, haemorrhagic stroke, and other types of stroke accounted for 68.81%, 26.69%, and 4.50%, respectively. Among total stroke cases, 59.47% were males and 40.53% were females; 66.99% were elderly and 33.01% were non-elderly; 48.59% occurred in the warm season, and 51.41% occurred in the cold season (Appendix A). The annual average levels of ambient NO_2_ during 2015–2018 were 18.96 μg/m^3^, 19.63 μg/m^3^, 23.41 μg/m^3,^ and 23.61 μg/m^3^, respectively (Appendix A). Compared with most regions worldwide, the levels of NO_2_ in the air were relatively low in Enshi city (Appendix A). The average daily temperature was 13.9 °C and ranged from −4.02 °C to 27.40 °C. The average daily temperature of the cold season was 7.65 °C and ranged from −4.02 °C to 21.19 °C, and the average daily temperature of the warm season was 20.10 °C and ranged from 5.69 °C to 27.40 °C (Appendix A). The average daily relative humidity was 78.62% and ranged from 3.44% to 97.40%.

Table 2 shows the Spearman correlation analysis results between air pollutants and meteorological factors in Enshi city. The results showed that the concentration of NO_2_ was significantly and positively correlated with PM_2.5_, PM_10_, SO_2,_ and CO (*p* < 0.01) but significantly and negatively correlated with O_3_ (8 h) (*p* < 0.01). The daily average temperature and relative humidity were also negatively correlated with NO_2_ (*p* < 0.01).

### 3.2. Exposure-Effect Analysis

Figure 2 describes the effects of NO_2_ exposure on the daily incidence of stroke and its subtypes in Enshi from 1 January 2015 to 31 December 2018. The single-day lag model results showed that NO_2_ exposure was associated with increased risk of incident total stroke with ER of 0.34% (95% CI: 0.002–0.67% for lag3) and 0.37% (95% CI: 0.04–0.70% for lag4), including the incident ischemic stroke (ER = 0.41%, 95% CI: 0.004–0.81% for lag3; ER = 0.58%, 95% CI: 0.18–0.98% for lag4). The cumulative lag model results showed that NO_2_ exposure was associated with increased risk of incident ischemic stroke with ER of 0.59% (95% CI: 0.01–1.17% for lag04) and 0.61% (95% CI: 0.01–1.21% for lag05). However, there were no statistically significant positive association between increased risk of incident haemorrhagic stroke or other types of stroke and ambient air NO_2_ levels.

### 3.3. Subgroup Analysis

Figure 3 summarises the results of the subgroup analysis. Changes in NO_2_ concentration were associated with the risk of incident stroke in males and elderly cases. The single-day lag effects of NO_2_ exposure on incident stroke in males (ER = 0.47%; 95% CI: 0.05–0.89% for lag4) and elderly (ER = 0.58%; 95% CI: 0.17–0.99% for lag3; ER = 0.59%; 95% CI: 0.19–0.99% for lag4) were statistically significant. The maximum cumulative lag effect of NO_2_ exposure on incident stroke in the elderly with ER of 0.75% (95% CI: 0.11–1.40% for lag07). No statistically significant association were observed in the subgroups of females and non-elderly cases. Another interesting finding was that a statistically significant positive association between incident stroke and NO_2_ was observed in the cold season, while a statistically significant negative association between incident stroke and NO_2_ was found in the warm season. In the cold season, the maximum single-day lag effect was 0.51% (95% CI: 0.03–0.99%) at lag1, and the maximum cumulative lag effect was 0.83% (95% CI: 0.15–1.51%) at lag05. In the warm season, the statistically significant negative association between incident stroke and NO_2_ was −0.75% (95% CI: −1.46–0.04%) at lag2.

### 3.4. Two-Pollution Model

Figure 4 presents the results of the two-pollution model. The associations between NO_2_ and incident stroke were statistically significant at some lags in two-pollution models. Furthermore, the cumulative lag effects were statistically significant on some lag days, including lag04 and lag05, after adjusting for PM_2.5_, PM_10,_ and O_3_. The single-day lag effects of NO_2_ were stronger after adjusting for other pollutants (PM_2.5_, PM_10,_ and SO_2_). For example, the percentage increase in the daily incidence of stroke was 0.43% (95% CI: 0.06–0.80%) for one unit (1 μg/m^3^) increase in NO_2_ after adjusting for SO_2_.

## 4. Discussion

NO_2_ is one of the main ambient air pollutants. Exposure to NO_2_ can lead to oxidative stress, vascular injury, blood viscosity, and other hazards, which may promote stroke development [28,29]. Previous studies proved that environmental NO_2_ exposure was associated with an increased risk of stroke in areas with severe air pollution [30,31,32]. Consistent with the abovementioned findings, our study aimed at the low air-pollution area and found that low-level ambient air NO_2_ exposure had short-term effects on the incident stroke. There was an association between short-term NO_2_ exposure and incident stroke in males and the elderly. Additionally, we observed statistically significant effects in the cold season. The results of our study provided evidence that short-time ambient air NO_2_ exposure can adversely influence the incident stroke, even at low pollution levels.

In the present study, we confirmed that the low-level NO_2_ exposure had short-term effects on the incident stroke, which was in line with previous studies. A pooled analysis of six European cohorts found long-term air-pollution exposure were associated with the incidence of stroke and coronary heart disease, even at pollutant concentrations lower than current limit values [22]. Another retrospective cross-sectional study in Tehran, where the pollution concentration was much higher than in Enshi city (the average annual level of NO_2_ was 59.0 ± 11.2 µg/m^3^), found that the level of air pollutants directly correlated with the number of stroke admissions to the emergency department [33]. However, inconsistent results have also been found in previous studies. A national cohort study in England found no statistically significant associations between the relatively low NO_2_ pollution levels and stroke, in which the annual average concentration of ambient air NO_2_ was 22.50 µg/m^3^ [34]. With regard to the reasons for these contradictions, the different compositions of the studied population and different geographical, racial, or ethnic factors in disparate regions might account for them [35,36]. After adjusting for another pollutant, the effects of NO_2_ remained significantly associated with incident stroke, confirming the independent effects of NO_2_ on the risk of incident stroke at low pollution levels. Previous studies found similar results, in which the effects of NO_2_ on stroke still existed after adjusting for another pollutant [37,38]. We also noted that, when adjusting for O_3_ in the two-pollutant model, the cumulative lag effects of NO_2_ on total stroke became enhanced and even statistically significant. This enhanced cumulative effect might be caused by the strong oxidability of O_3_, which could oxidise nitrogen oxide to create NO_2_ [39].

We noted that, in Enshi city, the low-level NO_2_ exposure had statistically significant effects on incident ischemic stroke but not haemorrhagic stroke. The heterogeneity might be explained by the underlying mechanisms of cardiovascular changes in relation to air pollution [37]. In vitro studies found that air-pollution exposure promoted vascular inflammation and lipid accumulation in foam cells, and accelerated the progression of atherosclerosis, all of which were typical prepathological changes in ischemic stroke [40,41]. These results were consistent with a 7 year time-series analysis in Korea, in which the investigators found a significant association between NO_2_ levels and the ischemic stroke mortality but not haemorrhagic stroke mortality [42]. However, several studies found statistically significant associations between NO_2_ and haemorrhagic stroke in areas with relatively higher air-pollution levels [37,43]. The different NO_2_ exposure levels in different studies might partly explain the difference. Air pollutants usually had a log-linear relationship with cardiovascular and cerebrovascular diseases, with a robust effect at low doses typically and a flattening of the dose-response at the highest exposure levels [44,45].

In this study, the subgroup analysis of age and sex revealed that males and the elderly were more susceptible to stroke when exposed to low-level ambient NO_2_ in the air. Similarly, a study about hospital admissions in China found that the elderly were more susceptible to stroke than the non-elderly [46]; another study found that male stroke patients were more susceptible to changes in NO_2_ concentrations than female patients [47]. However, some previous studies found different results regarding the susceptibility of age to stroke. A case-crossover study conducted in seven cities in Australia and New Zealand showed that elderly people were not susceptible to stroke [48]. The different susceptibility of people of different sexes and ages might be related to their physiological characteristics and lifestyle. Elder individuals with weak immune systems and chronic underlying diseases might be more sensitive to air-pollution exposure [49,50].

In Enshi city, the association between NO_2_ exposure and incidence of total stroke was found significantly positive in the cold season but was not positive in the warm season. These results were consistent with those obtained by a study conducted in Wuhan, another city in Hubei, China, which found that NO_2_ exposure in the cold season was significantly associated with stroke hospitalisation [51]. The positive associations might be explained by the special biologically plausible evidence in the cold season. Previous studies reported that low temperature was related to an increase in inflammatory response level, the oxidative stress level in brain tissue, blood pressure, and autonomic nervous system changes [52,53]. All of these changes were related to the occurrence and development of stroke. The different susceptibility to stroke between cold and warm seasons might be explained by the different human activity patterns in different seasons. For example, the use of air conditioning and coal combustion was more frequent in the cold season but not in the warm season.

This study also had some limitations. Firstly, this study might underestimate the effects of NO_2_ on incident stroke, because the data of ambient air pollutants and meteorological factors (obtained from fixed monitoring stations) could not represent total exposure to the population. Secondly, the meteorological factors, such as atmospheric pressure and wind speed, were not included in our GAM models, because no evidence had been found to support their direct connections. In addition, we failed to obtain case data, including individual air-pollution data, exercise, diet, smoking, and other information. We would carry out further relevant research and improve the conclusion.

## 5. Conclusions

Through this time-series analysis, we found that short-term exposure to low-level NO_2_ increased the risk of incident stroke, including ischemic stroke, in Enshi city. Furthermore, males and the elderly might be susceptible to stroke when exposed to low-level NO_2_, and the adverse effects in the cold season were more remarkable. These results provided evidence to inform policymaking aimed at protecting public health from air pollution, especially in regions with low-level air pollution.

## Figures and Tables

**Figure 1 ijerph-19-06683-f001:**
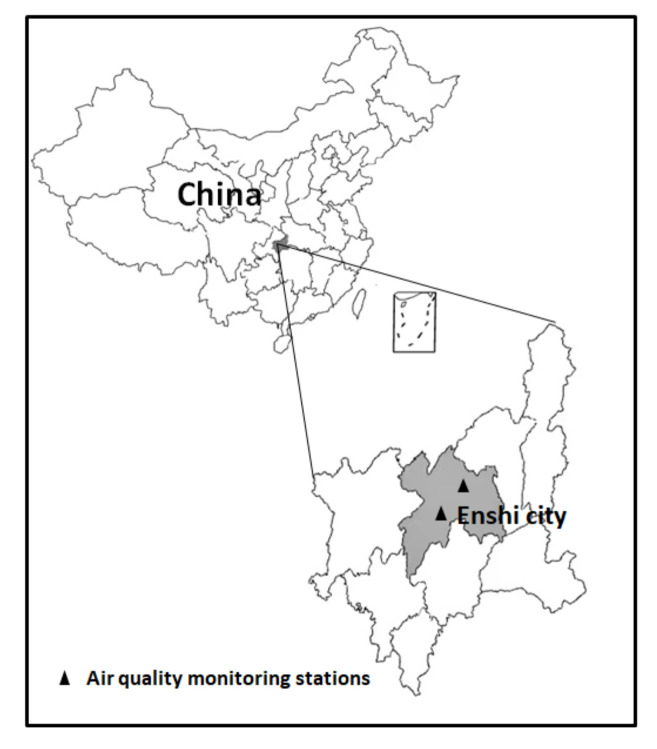
Map of Enshi and the location of national air quality monitoring stations.

**Figure 2 ijerph-19-06683-f002:**
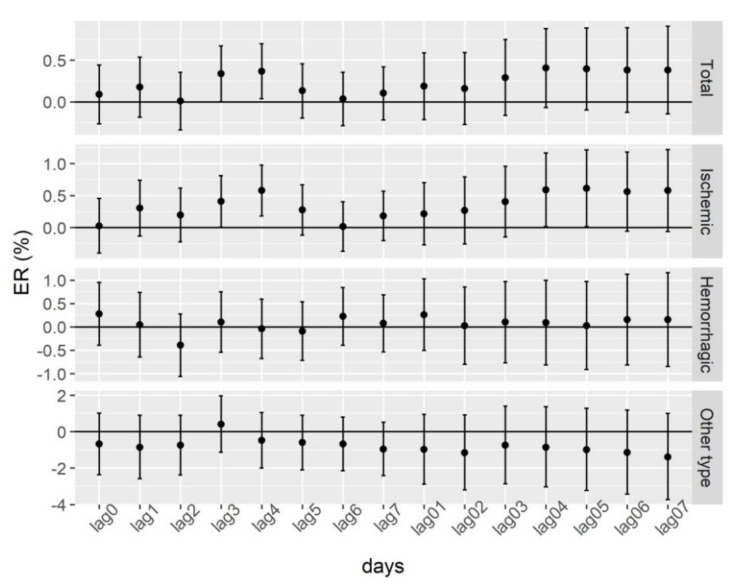
ER and 95% CI of daily incidence of different subtypes of stroke associated with one-unit (1 μg/m^3^) increase in NO_2_. Single-day lag effects: lag0, lag1, lag2, lag3, lag4, lag5, lag6, and lag7 (lag0 corresponds to the current-day air-pollutant concentration, and lag1 refers to the average concentration of air pollutants on the day before stroke incidence, etc.). Cumulative lag effects: lag01, lag02, lag03, lag04, lag05, lag06, and lag07 (the lag01 corresponds to the 2-day moving average concentration of air pollutants concentrations on the present day and previous day, etc.).

**Figure 3 ijerph-19-06683-f003:**
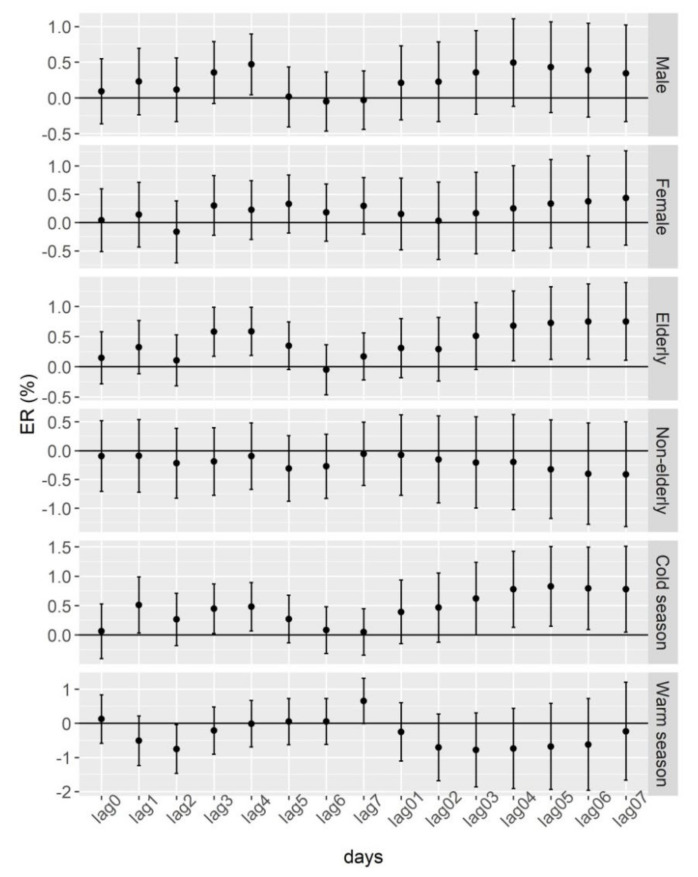
Associations of air NO_2_ with daily incident stroke in different subgroups. ER: the percentage of increase in the daily incidence of stroke associated with one-unit (1 μg/m^3^) increase in NO_2_. Single-day lag effects: lag0, lag1, lag2, lag3, lag4, lag5, lag6, and lag7 (lag0 corresponds to the current-day air-pollutant concentration, and lag1 refers to the average concentration of air pollutants on the day before stroke incidence, etc.). Cumulative lag effects: lag01, lag02, lag03, lag04, lag05, lag06, and lag07 (the lag01 corresponds to the 2-day moving average concentration of air-pollutant concentrations on the present day and previous day, etc.).

**Figure 4 ijerph-19-06683-f004:**
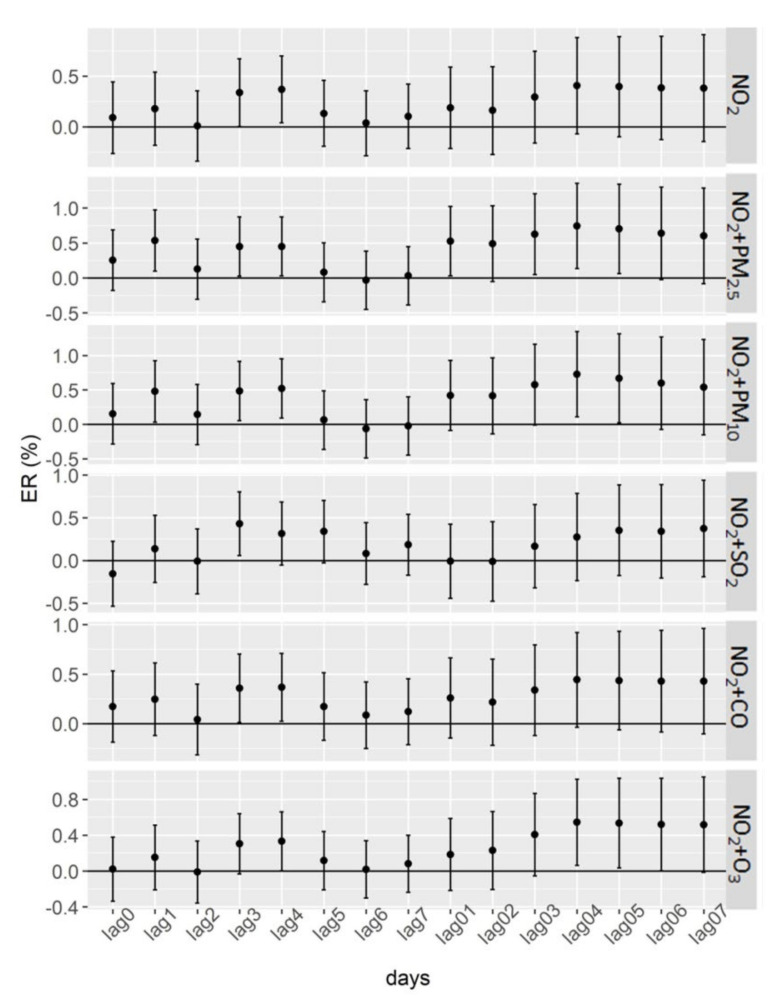
Associations of NO_2_ with the daily incidence of stroke after adjusting for the interference effects of other air pollution. ER: the percentage increase in the daily incidence of stroke associated with one-unit (1 μg/m^3^) increase in NO_2_. Single-day lag effects: lag0, lag1, lag2, lag3, lag4, lag5, lag6, and lag7 (lag0 corresponds to the current-day air-pollutant concentration, and lag1 refers to the average concentration of air pollutants on the day before stroke incidence, etc.). Cumulative lag effects: lag01, lag02, lag03, lag04, lag05, lag06, and lag07 (lag01 corresponded to the 2-day moving average concentration of air pollutants concentrations on the present day and previous day, etc.).

**Table 1 ijerph-19-06683-t001:** Descriptive statistics for daily incidence of stroke, air pollutant concentrations, and meteorological factors in Enshi city, 2015–2018.

Variables	Mean ± SD	Min	P_25_	Median	P_75_	Max
Daily incidence of stroke						
Total	6.25 ± 3.32	0	4	6	8	19
Ischemic	4.30 ± 2.71	0	2	4	6	15
Haemorrhagic	1.67 ± 1.43	0	1	4	2	7
Other types ^1^	0.28 ± 0.56	0	0	0	0	4
Elderly ^2^	4.18 ± 2.53	0	2	4	6	15
Non-elderly ^3^	2.06 ± 1.64	0	1	2	3	13
Male	3.71 ± 2.30	0	2	3	5	13
Female	2.53 ± 1.88	0	1	2	4	10
Warm season ^4^	6.06 ± 3.10	0	4	6	8	19
Cold season ^5^	6.44 ± 3.53	0	4	6	8	19
Air pollutants						
PM_2.5_ (µg/m^3^)	45.98 ± 30.12	9.16	25.08	36.96	56.08	248.96
PM_10_ (µg/m^3^)	67.94 ± 38.14	16.60	41.69	57.29	82.86	292.79
SO_2_ (µg/m^3^)	9.68 ± 7.75	1.58	5.83	7.75	11.52	61.68
NO_2_ (µg/m^3^)	21.40 ± 10.21	3.44	14.34	19.73	27.31	72.94
CO (mg/m^3^)	0.88 ± 0.34	0.20	0.64	0.85	1.03	2.53
O_3_ (µg/m^3^)	61.21 ± 31.51	2.63	38.62	58.94	81.83	184.31
Meteorological factors						
Temperature (°C)	13.89 ± 7.64	−4.02	6.98	14.57	20.62	27.40
Humidity (%)	78.62 ± 9.48	42.71	71.80	79.64	86.74	94.40

^1^ Other types are strokes that do not specifically refer to hemorrhage or ischemia. ^2^ The age ≥ 65 years were defined as elderly group. ^3^ The age < 65 years were defined as non-elderly. ^4^ Warm season: April, May, June, July, August, and September. ^5^ Cold season: October, November, January, December, January, February, and March.

**Table 2 ijerph-19-06683-t002:** Spearman correlation coefficients between daily air-pollutant concentrations and meteorological factors in Enshi city, 2015–2018.

Variables	PM_2.5_	PM_10_	SO_2_	NO_2_	CO	O_3_	Temperature	Humidity
PM_2.5_	1.000							
PM_10_	0.943 **	1.000						
SO_2_	0.527 **	0.474 **	1.000					
NO_2_	0.556 **	0.599 **	0.351 **	1.000				
CO	0.255 **	0.199 **	0.095 **	0.372 **	1.000			
O_3_	−0.296 **	−0.223 **	−0.141 **	−0.398 **	−0.264 **	1.000		
Temperature	−0.491 **	−0.429 **	−0.161 **	−0.516 **	−0.188 **	0.615 **	1.000	
Humidity	−0.258 **	0.361 **	−0.198 **	−0.073 **	0.208 **	−0.436 **	−0.056 *	1.000

* *p* < 0.05; ** *p* < 0.01.

## Data Availability

Restrictions apply to the availability of these data. Data were obtained from the China Disease Prevention and Control Information System.

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
