# Peer review of "Short-Term Effects of Low-Level Ambient Air NO_2_ on the Risk of Incident Stroke in Enshi City, China"

_ijerph, 2022, doi:10.3390/ijerph19116683_

Round 1
Reviewer 1 Report
This study aimed to investigate the effect of the short-term effects of NO2 on the risk of stroke in a relatively low-pollution area, Enshi city of Hubei, China. The study used generalised additive model (GAM) and statistical analysis to assess the effect and relationship between the studied variables. This research has originality. However, it is necessary for the authors to clearly specify the research contribution. In general, it is an interesting study, but several issues need to be addressed as follows:
The abstract should follow the journal format. The authors combined subtopics in one paragraph, for example, the titles namely Background, Method, Results, and Conclusions, might not be necessary for this template.
More information about the GAM model should be provided in detail.
Please provide more information about the stroke case and sample size selection. How the study controls the background of stroke cases? Do other environmental impacts or activities have no effect the stroke?
The authors should provide the location of the city, the boundary of the studied area, and the location of 2 national IAQ monitoring stations.
The specification of the use of Equation 1 should be clearly defined in terms of specification, limitation, and its usage. How this model can accurately predict the risk of short-term NO2 exposure with a low concentration?
What are Lag0 and Lag7 represent? It should be provided in detail.
Please provide more detailed information about the sensitivity analysis. For example, which model was used, and how many simulation cases were involved?
Table S1 and Table S2 are missing?
In, Figure 1, Figure 2, and Figure 3, the ER shows in %. It is not sure whether the numbers shown on the y-axis represent a scale of 0-1 or the percentage value.
The conclusions section underscores the scientific value and the applicability of the study’s results. What are the research gaps/contributions? What is the study objective, and how was the data analyzed?
Author Response
Point 1: The abstract should follow the journal format. The authors combined subtopics in one paragraph, for example, the titles namely Background, Method, Results, and Conclusions, might not be necessary for this template.
Response 1: We appreciate the reviewer’s suggestion. We have corrected the abstract in the revised manuscript to follow the journal format.
Point 2: More information about the GAM model should be provided in detail.
Response 2: Thank you for the suggestion. The detailed description of the GAM model, including equations of the single-pollutant model and the two-pollutant model has been added to the revised manuscript (see Materials and Methods, Statistical analysis section).
Point 3: Please provide more information about the stroke case and sample size selection. How the study controls the background of stroke cases? Do other environmental impacts or activities have no effect the stroke?
Response 3: We appreciate the reviewer’s comments.
1) In this time-series study, the information of stroke cases collection is as follows. All the stroke cases were clinical diagnosed by the local sentinel hospitals and simultaneously reported to the China Disease Prevention and Control Information System through case report cards. Through the system, we downloaded the data of all reported incident stroke cases, including the ID number, sex, age, date of birth, onset date of stroke, type of stroke diagnosis, ICD-10 encoding and current address for each case. The annual size of the reported incident stroke cases ranged from 1448 to 2686 between 2015 and 2018 in Enshi city. These details have been added to the revised manuscript (see Materials and Methods, Data sources section).
2) We controlled the background of stroke cases by performing different GAM models and subgroup analysis. In the GAM models, confounding factors, including environmental (such as mean temperature and relative humidity) and air quality parameters (including PM2.5, PM10, SO2, CO and O3) were included simultaneously or separately; in the subgroup analysis, the basic characteristics of stroke cases (including ages and sexes) and onset seasons were considered. The above analysis should help control the background of stroke cases.
3) According to the results of previous studies, individual activities, such as smoking, substance abuse and physical inactivity, might be contributing factors to stroke [1, 2]. Limited by the China Disease Prevention and Control Information System, the individual activity information of stroke cases were not collected, and thus not analyzed in this study. However, the basic individual characteristics of cases including case genders, ages and onset seasons were analyzed through the subgroup analysis.
Point 4: The authors should provide the location of the city, the boundary of the studied area, and the location of 2 national IAQ monitoring stations.
Response 4: We appreciate this suggestion. The location of Enshi city and the 2 IAQ monitoring stations has been added to the revised manuscript (see Materials and Methods, Data sources, Figure 1).
Point 5: The specification of the use of Equation 1 should be clearly defined in terms of specification, limitation, and its usage. How this model can accurately predict the risk of short-term NO2 exposure with a low concentration?
Response 5: We appreciate the reviewer’s comments.
The specification of the use of Equation 1 has been added to the revised manuscript (see Materials and Methods, Statistical analysis section). According to Ravindra K, a GAM is the classical addendum of general linear models, which provides a structure for generalizing a general linear model by allowing additivity of non-linear functions of the variables [3]. The specification, limitation and usage of GAM can be found in Ravindra K’s report, and we have cited this literature in the article.
In time-series studies, the GAM analysis should be suitable to evaluate the risk of short-term air pollutant exposure, including pollutants with relatively low concentrations. Liu et al. used GAM model to analysis short-term effect of relatively low level air pollution on outpatient visit in China [4]; a pooled analysis of six European cohorts within the ELAPSE project also used the GAM model to evaluate the associations between long-term exposure to low-level ambient air pollution and incidence of stroke and coronary heart disease [5].
Point 6: What are Lag0 and Lag7 represent? It should be provided in detail.
Responses 6: Thanks for the comments. Lag0 corresponded to the current-day air pollutant concentration and lag1 referred to the average concentration of air pollutants in the day before incidence stroke and so on. We have added the descriptions in the revised manuscript (see Materials and Methods, Statistical analysis section).
Point 7: Please provide more detailed information about the sensitivity analysis. For example, which model was used, and how many simulation cases were involved?
Responses 7: Thanks for these comments. We have added the description of sensitivity analysis in the revised manuscript (see Materials and Methods, Statistical analysis section).
Point 8: Table S1 and Table S2 are missing?
Responses 8: The Table S1 and Table S2 have been uploaded to the submission system as the supplementary materials.
Point 9: In, Figure 1, Figure 2, and Figure 3, the ER shows in %. It is not sure whether the numbers shown on the y-axis represent a scale of 0-1 or the percentage value.
Responses 9: Thanks for the comment. In Figure 1, Figure 2 and Figure 3, the numbers shown on the y-axis represent the percentage value.
Point 10: The conclusions section underscores the scientific value and the applicability of the study’s results. What are the research gaps/contributions? What is the study objective, and how was the data analyzed?
Responses 10: Thanks for these comments. The contributions of this study were to inform policy making aimed at protecting public health from air pollution, especially in regions with low-level air pollution. The gaps of this study were failed to obtain individual data of the cases, including individual exposure levels to air pollutants , exercise, diet and smoking information, and the data of ambient air pollutants and meteorological factors (obtained from fixed monitoring stations) could not represent total exposure to the population (see Discussion section). The objective of this study was to assesse the short-term effects of low-level ambient air NO2 exposure on the risk of incident stroke in Enshi city, through a time-series analysis using GAM models. We have revised the conclusion section accordingly in the revised manuscript (see Conclusion section).
References
- Meschia JF, Bushnell C, Boden-Albala B, et al. Guidelines for the primary prevention of stroke: a statement for healthcare professionals from the American Heart Association/American Stroke Association. Stroke. 2014;45(12):3754-3832. doi:10.1161/STR.0000000000000046.
- Roswall N, Pyko A, Ögren M, et al. Long-Term Exposure to Transportation Noise and Risk of Incident Stroke: A Pooled Study of Nine Scandinavian Cohorts. Environ Health Perspect. 2021;129(10):107002. doi:10.1289/EHP8949.
- Ravindra K, Rattan P, Mor S, Aggarwal AN. Generalized additive models: Building evidence of air pollution, climate change and human health. Environ Int. 2019 Nov;132:104987. doi: 10.1016/j.envint.2019.104987. Epub 2019 Aug 6. PMID: 31398655.
- 17.18. Liu C. C., Liu Y. W., Zhou Y. D., et al. Short-term effect of relatively low level air pollution on outpatient visit in Shennongjia, China. Environ. Pollut. 2019, 245, 419-426. doi:10.1016/j.envpol.2018.10.120.
- 21. Wolf K., Hoffmann B., Andersen Z. J., et al. Long-term exposure to low-level ambient air pollution and incidence of stroke and coronary heart disease: a pooled analysis of six European cohorts within the ELAPSE project. The Lancet. Planetary health 2021, 5, (9), e620-e632. doi:10.1016/s2542-5196(21)00195-9.

Reviewer 2 Report
Abstract
The authors need to restate this finding: “the largest significant estimate effect of NO2 on total incident stroke was 0.37%” Percent of what exactly? The term “lag 4” is also not defined. 0.37% seems like a very small effect (under 1%), but I don’t think this was the authors’ intention.
The abstract describes “The two-pollutant model” but only one pollutant (NO2) is mentioned.
It is also not clear exactly what the p-values are comparing.
The abstract does not state how other potential causes of stroke were controlled for in this study, or if confounding factors were considered at all.
“Low level” does not seem like an appropriate key word
Introduction
Is stroke still one of the leading causes of poor health in this time of covid?
NO2 also has indoor sources such as gas stoves or smoking, but this article does not mention any indoor sources at all.
Methods
Line 126. The term “two-pollutant model” could be defined here.
Results
Percentages of what exactly? Needs better definition and perhaps correction of decimal point?
Conclusion
Line 299. What exactly does it mean to “pay attention to environmental pollution control”?
The English in this paper is substandard and needs improvement
Author Response
Point 1: The authors need to restate this finding: “the largest significant estimate effect of NO2 on total incident stroke was 0.37%” Percent of what exactly? The term “lag 4” is also not defined. 0.37% seems like a very small effect (under 1%), but I don’t think this was the authors’ intention.
Responses: Thanks for these comments.
1) In this study, we reported the risk estimation as excess risk (ER), defined as the percentage increase in daily incidence of stroke with one unit (1 μg/m3) change of NO2. We have corrected the description “the largest significant estimate effect of NO2 on total incident stroke was 0.37%” as follows: “We found that NO2 had statistically significant effects on incidence of stroke and ischemic stroke, estimated by excess risks (ER) were 0.37% (95% CI, 0.04% - 0.70%) and 0.58% (95% CI, 0.18% - 0.98%), respectively.”
2) The term “lag 4” corresponded to the average concentration of air pollutants in the fourth day before incidence stroke. And we have added the description of the lag in the revised manuscript (see Materials and Methods, Statistical analysis section).
3)As described above, the excess risk (ER) value means the percentage increase in daily incidence of stroke with one unit change of NO2 in this study. In this study, the unit of NO2 was 1 μg/m3 when ER was calculated. According to previous studies, the ER value of 0.37% was not a very small effect. For example, according to Zhang Y , a 10 µg/m³ increase of ambient NO2 was associated with 2.07% (95% CI: 1.08% - 3.07%) excess risk, which meant that a 1 µg/m³ increase of ambient NO2 was associated with 0.207% (95% CI: 0.108% - 0.307%) excess risk [6].
Point 2: The abstract describes “The two-pollutant model” but only one pollutant (NO2) is mentioned.
Responses 2: Thanks for the comment. We have added the results of two-pollutant model into the abstract section of the revised manuscript.
Point 3: It is also not clear exactly what the p-values are comparing.
Responses 3: Thanks for the comment. The P-values have been removed from the manuscript.
Point 4: The abstract does not state how other potential causes of stroke were controlled for in this study, or if confounding factors were considered at all.
Responses 4: Thanks for the comment. In this study, potential confounding factors, such as public holidays (Holiday), day of the week (DOW), long-term trends (Time) and meteorological factors (temperature and relative humidity), were included in the GAM analysis as confounding factors. We have added the information to the abstract (see Abstract section).
Point 5: “Low level” does not seem like an appropriate key word
Responses 5: We appreciate the reviewer’s comment. We have removed “Low level” from the keywords in revised manuscript.
Introduction
Point 6: Is stroke still one of the leading causes of poor health in this time of covid?
Responses 6: We understand the reviewer’s concern. According to Douiri A’, during the COVID-19 pandemic, stroke remains one of the important causes of poor health worldwide [7]. We have corrected the description in the revised manuscript.
Point 7: NO2 also has indoor sources such as gas stoves or smoking, but this article does not mention any indoor sources at all.
Responses 7: Thanks for the comment. We have added the indoor sources of NO2 into the introduction section of the revised manuscript. (see Introduction section)
Methods
Point 8: Line 126. The term “two-pollutant model” could be defined here.
Responses 8: Thanks for these comments. We added the description of two-pollutant model into the statistical analysis section of the revised manuscript.
Results
Point 9: Percentages of what exactly? Needs better definition and perhaps correction of decimal point?
Responses 9: Thanks for the comment. “Percentages” refers to corresponding percentage increase in daily incidence of stroke for per 1 μg/m3 increase of NO2. We have corrected the descriptions in the revised manuscript.
Conclusion
Point 10: Line 299. What exactly does it mean to “pay attention to environmental pollution control”?
Responses 10: We understand the reviewer’s concern. The sentence “pay attention to environmental pollution control” means that more concern on the health hazards of air pollutants should be pay. We have revised the discussion section of the revised manuscript.
Point 11: The English in this paper is substandard and needs improvement
Responses 11: Thanks for the comment. We have asked colleague who is native English speaker to correct the details in this manuscript.
References
- Zhang Y, Wu K, Zhu C, Feng R, Li C, Ma L. [Association between ambient air pollution and stroke mortality in Wuhan, China: A time-series analysis]. Zhonghua Yu Fang Yi Xue Za Zhi. 2015 Jul;49(7):605-10. Chinese. PMID: 26310472.
- Douiri A, Muruet W, Bhalla A, James M, Paley L, Stanley K, Rudd AG, Wolfe CDA, Bray BD; SSNAP Collaboration. Stroke Care in the United Kingdom During the COVID-19 Pandemic. Stroke. 2021 Jun;52(6):2125-2133. doi: 10.1161/STROKEAHA.120.032253. Epub 2021 Apr 26. PMID: 33896223; PMCID: PMC8140645.

Reviewer 3 Report
ijerph-1700994-Short-term effects of low-level ambient air NO2 on the risk of 2 incident stroke in Enshi city, China
General comments
This manuscript is examined the associations between low-level NO2 exposure and the risk of incident stroke, taking into consideration the stroke sub-types in China. Stroke cases were defined by the data from China Disease Prevention and Control Information System through case report cards. The daily air pollutant data were collected from two national air quality control automatic monitoring stations and Hubei University for Nationalities monitoring station, from 1st January 2015 to 31th December 2018. Subgroup analysis was also conducted according to sex, age and season to demonstrate susceptibility in different subgroups and seasons. The data and study design are robust, and manuscript is well-written even though several information was missing, especially, in current version of the manuscript, the interpretation of the results from the figures is difficult to understand for the readers who are not familiar with statistical analyses, such as lag effects. In addition, I have seen that there are some points can be improved before publishing.
Minor comments
- L134-141, It would be more informative if the demographic information described in the manuscript L134-141 such as numbers of incident stroke cases, sex ratio, and age were shown in Supplementally table. Please add them in supplemental.
- L140, The annual average levels 139 of ambient NO2 during 2015-2018 were 18.96 μg/m3, 19.63 μg/m3, 23.41 μg/m3…., These information also should be shown in the table. (Maybe as Table S2?)
- L145, Authors mentioned Table S2, however, I could not find it in supplemental file.
- Figure 1, Maybe because I am not familiar with “lag”, but it would be informative for the readers if the meanings/interpretations of lag, and also lag0, lag1,lag7, lag01 ….lag07 are explained in the footnote of the figure.
- L164-173, I have seen that statistically significance was shown in very limited points, such as ischemic stroke at lag 04 (marginal significance?) and lag 05.
- Figure 2, same as figure 1, what does lag4 in male represent? Please add explanation for it in the methods/statistical analysis section.
- Figure 2 and L261-273, It is reasonable and clearly shown that the elderly person and cold season are more susceptible than non-elderly and summer season, however, for the sex, I could not see those clear differences between male and female. Are there any interaction effect of the sex? Because only one point in the male at lag4 was statistically significant, I recommend that the statement regarding sex difference tone down.
End

Author Response
Point 1: L134-141, It would be more informative if the demographic information described in the manuscript L134-141 such as numbers of incident stroke cases, sex ratio, and age were shown in Supplementally table. Please add them in supplemental.
Responses 1: Thanks for these comments. We have added the demographic information in supplemental table S3. The content of the table is as follows:
Table S3. The demographic information for incidence of stroke in Enshi city, 2015-2018
|
Classification |
Incident stroke numbers (%) |
|
Total |
9122(100) |
|
Stroke subtypes |
|
|
Ischemic |
6277(68.81) |
|
Hemorrhagic |
2435(26.69) |
|
Other types 1 |
410(4.50) |
|
Ages |
|
|
Elderly 2 |
6111(66.99) |
|
Non-elderly 3 |
3011(33.01) |
|
Sexes |
|
|
Male |
5425(59.47) |
|
Female |
3697(40.53) |
|
Seasons |
|
|
Warm season 4 |
4432(48.59) |
|
Cold season 5 |
4690(51.41) |
1 Other types are strokes that do not specifically refer to bleeding or infarction. 2 The age ≥ 65 years be defined as elderly group. 3 The age < 65 years be defined as non-elderly. 4 Warm season: April, May, June, July, August and September. 5 Cold season: October, November, January, December, January, February and March.
Point 2: L140, The annual average levels 139 of ambient NO2 during 2015-2018 were 18.96 μg/m3, 19.63 μg/m3, 23.41 μg/m3…., these information also should be shown in the table. (Maybe as Table S2?)
Responses 2: Thanks for these comments. We have added the annual average NO2 levels in table S4 of the supplemental materials. The contents of table S4 is as follows:
Table S4. The annual average levels of ambient NO2 and incident stroke cases in Enshi city during 2015-2018
|
Variables |
2015 |
2016 |
2017 |
2018 |
2015-2018 |
|
Annual average(µg/m3) |
18.96 |
19.63 |
23.41 |
23.61 |
21.40 |
|
Annual incident stroke cases |
2683 |
2433 |
1448 |
2558 |
9122 |
Point 3: L145, Authors mentioned Table S2, however, I could not find it in supplemental file.
Figure 1, Maybe because I am not familiar with “lag”, but it would be informative for the readers if the meanings/interpretations of lag, and also lag0, lag1, lag7, lag01 ….lag07 are explained in the footnote of the figure.
Responses 3: Thanks for these comments. The contents of Table S2 have been uploaded to the submission system in as the supplementary materials. And we have defined the lag in the revised manuscript (see Materials and Methods, Statistical analysis section) and added the meanings of lag0, lag1, lag7, lag01 ….lag07 in the footnote of the figures.
Point 4: L164-173, I have seen that statistically significance was shown in very limited points, such as ischemic stroke at lag 04 (marginal significance?) and lag 05.
Responses 4: We understand the reviewer’s concern. There was marginally significant, such as ischemic stroke at lag04 (ER= 0.59%; 95% CI, 0.01% - 1.17%) and lag05 (ER= 0.61%; 95% CI, 0.01% - 1.21%), and there was non-marginally significant about ischemic stroke at lag4 (ER = 0.58%; 95% CI, 0.18% - 0.98%). We have added the ER values of lag04 and lag05 to the revised manuscript (see Results, Exposure - effect analysis section).
Point 5: Figure 2, same as figure 1, what does lag4 in male represent? Please add explanation for it in the methods/statistical analysis section.
Responses 5: Thanks for thee comment. The lag4 in male represented the effect of air pollutants in the fourth day before stroke incidence on incident stroke for males. And we have added the description of the lag in the Statistical analysis section, as follows:
In single-day exposure models the lag0 corresponded to the current-day air pollutant concentration and lag1 referred to the average concentration of air pollutants in the day before incidence stroke and so on. In cumulative exposure models the lag01 corresponded to the 2-day moving average concentration of air pollutants concentrations on the present day and previous day and so on.
Point 6: Figure 2 and L261-273, It is reasonable and clearly shown that the elderly person and cold season are more susceptible than non-elderly and summer season, however, for the sex, I could not see those clear differences between male and female. Are there any interaction effect of the sex? Because only one point in the male at lag4 was statistically significant, I recommend that the statement regarding sex difference tone down.
Responses 6: Thanks very much for the suggestion. In this study, we explored the possible sex differences behind the effects of NO2 exposure on incidence of stroke, and there were not any interaction effect of the sexes. The statements regarding sex differences have be tone down in the revised manuscript (see the Discussion).
The revised manuscript in the attachment.

Round 2
Reviewer 2 Report
The revised version is suitable for publication. Thank you for the opportunity to review it